# Spatial Effects of Railway Network Construction on Urban Sprawl and Its Mechanisms: Evidence from Yangtze River Delta Urban Agglomeration, China

**Yuan Yi** **, Fang He and Yuxuan Si \***

School of Economics and Management, Tongji University, Shanghai 200092, China; 1810045@tongji.edu.cn (Y.Y.); heyoufang@tongji.edu.cn (F.H.)
* Correspondence: 2310123@tongji.edu.cn; Tel.: +86-180-1749-6839

**Abstract:** Urban sprawl has become a notable feature in China. Previous studies have found that railway development has a significant effect on urban sprawl. However, the detailed mechanisms of how railways affect urban sprawl have not been studied in depth. Furthermore, China's railway system has already formed a network. The network status of cities within the railway network may affect urban sprawl, but few studies have examined this factor. In this context, to explore the effects of railway networks on urban sprawl and the mechanisms of these effects, this study applied the social network analysis (SNA) method to measure the indicators of railway network characteristics and conducted panel model regression with the above indicators using the data from 26 cities from 2011 to 2019 in the Yangtze River Delta (YRD) in China. The main conclusions are as follows: (1) Railway network construction has a significantly positive impact on urban sprawl through the network agglomeration and diffusion mechanisms. (2) The network agglomeration mechanism improves the location condition of the central cities on the railway network, which encourages urban sprawl as an agglomeration pattern. (3) The network diffusion mechanism enhances the integration of the peripheral cities with the central city on the railway network, which encourages urban sprawl as a diffusion pattern. The network diffusion mechanism is heterogeneous in metropolitan areas (MAs) for the different levels of central city agglomeration. The findings provide a reference for railway construction and urban planning.

**Keywords:** spatial effects; railway network; urban sprawl; central cities; peripheral cities; social network analysis; panel regression model

## 1. Introduction

China's economy has developed rapidly since the reform and opening up in 1978. Since the 1980s, urbanization in China has entered a stable and rapid growth phase, and urban sprawl has become a notable feature of this process. With a huge amount of the population moving from rural to urban areas, a large amount of rural land was converted into urban land. Over the period of 1990–2022, China's urban built-up areas increased from 12,900 to 63,700 square kilometers, which represents a four-fold increase in nearly 40 years. Additionally, during the period of 2013–2022, China's urban built-up areas increased by 33%, even exceeding the growth of the urban population, which was 30.5% during the same period [1]. However, this rapid urban sprawl has also led to problems such as conflicts of interests in land expropriation [2,3], inefficient use of and idle land [4], and ecological destruction [5,6]. Therefore, urban sprawl has drawn widespread attention. It is critical to explore the determining drivers and mechanisms of urban sprawl, within the context of Chinese authorities urging for efforts to boost the high-quality development of urbanization.

Transport infrastructure plays a vital role in urban sprawl [7]. In developing countries such as China, the railway network is an important transport infrastructure, which

has a significant impact on the land use–cover change (LUCC) and the spatial layout of cities [8]. On the one hand, the construction of a railway network can improve the accessibility of a city, which encourages the agglomeration of industries and labor [9]. Due to spatial–temporal effects [10], a railway network significantly shortens the travel time between locations, improves the accessibility of areas along railway lines, and enhances the connectivity among regions. On the other hand, railway networks can intensify integration among cities, which encourages the diffusion of resources and factors. Owing to the spatial spillover effect, railways strengthen the economic linkages among cities [8,11] and promote the spatial flow of population, goods, information, capital, and technology [12–14].

In recent years, the Chinese government has made continued efforts to promote the construction of the railway network given its benefits for economic growth and urbanization. Authorities have asked to accelerate the construction of intercity railway networks in key urban agglomerations in order to strengthen the radiation and driving force of central cities into peripheral cities [15]. With the continuous improvement in the railway network, it is expected that it will play a more critical role in the construction and development of cities. Therefore, it is of theoretical and practical significance to explore the spatial effects of a railway network on urban sprawl.

Although previous research has studied the impact of railways on urban sprawl, most of it has been limited to examining the significance of the effects. There is a lack of in-depth studies analyzing the mechanisms in detail, especially from the theoretical perspective of agglomeration and diffusion mechanisms. For study methods, the existing literature usually uses the differences-in-differences (DID) or multistage DID model with policy implementation as the treatment to identify differences in urban sprawl before and after the opening of railways [16]. Since the opening of railways in a certain period is continuous and random in practice, that is, policy implementations not only take place over specific years, the DID or multistage DID model cannot depict the changes in railway interactions among objects well. Instead, social network analysis (SNA), one of the mainstream methods in complex network study areas, can be used to accurately measure the accessibility and strength of connectivity among the objects in the same network, although few researchers have applied SNA to study railway networks.

To bridge this gap, this study mainly conducted three bodies of work, as follows: (1) First, we constructed an analytical framework on the basis of agglomeration and diffusion theory and proposed two main mechanism hypotheses regarding the impact of railways on urban sprawl—the network agglomeration mechanism and the network diffusion mechanism. (2) Second, with a dataset from 26 cities in the Yangtze River Delta (YRD) in China from 2011 to 2019, we applied SNA to measure the railway network linkage characteristics among cities in the YRD through the network centrality indicator and the connection strength indicator. (3) Third, we constructed fixed-effect panel regression models with the network centrality indicator as a proxy variable for location condition and the connection strength indicator as a proxy variable for the degree of integration to further verify the two main hypotheses on urban sprawl. Our findings are expected to promote high-quality development of railway networks and the effective use of land resources.

The rest of this paper is organized as follows: Section 2 provides a literature review. Section 3 presents the research hypotheses. Section 4 introduces the materials and methods. Section 5 reports the empirical results. A discussion is presented in Section 6. Finally, Section 7 draws the conclusions and implications.

## 2. Literature Review

### 2.1. Urban Sprawl and Its Drivers

Urban sprawl is a certain result of economic development and urbanization, which is currently one of the most prominent features of LUCC. The patterns of urban sprawl mainly include infill, extension, expansion, and linear development [17]. In addition, urban sprawl shows spatial correlations and hierarchical differences, and the patterns and drivers of urban sprawl vary widely among cities in different regions at different development

levels [18,19]. In China, the rate of urban sprawl is faster in the eastern coastal regions than in the western and central regions, and it is faster in prefecture-level cities than in county-level cities. Moreover, the larger the size of a city, the faster the rate of urban sprawl [20].

Scholars have studied the impact of urban sprawl on social, economic, and sustainable eco-environmental development [21]. They found that urban–rural income disparities are significantly related to rapid urban expansion in developing countries, and there is an inverted U-shaped relationship between them in China [22]. The inverted U-shaped relationship was also found between urban size and regional economic integration. In addition, researchers found that urban sprawl in China has increased carbon dioxide emissions from transportation, construction, and industry [23] and significantly reduced eco-environmental quality [24].

The literature on the drivers of urban sprawl has mainly focused on natural resource endowment [25], economy [26], population [27], infrastructure construction [28], and policy [29]. Among these, natural resource endowment plays a fundamental role. Site conditions determine the potential for regional economic development, which has a direct impact on changes in urban land use. Geological conditions and environmental pressures can limit urban sprawl [30]. Economic development and population growth are the direct and fundamental drivers of urban sprawl [27]: economic growth attracts, for example, capital, industry, and public services to the urban area and increases the income of residents, leading to increased demand for land [31,32]. Social mobility and rural–urban migration increase the urban population, leading to urban sprawl [33]. In addition, political factors cannot be ignored. Local governments' reliance on land financing has strongly encouraged urban sprawl in recent years [34]. Meanwhile, the central government's policy of building land quotas has partly limited expansion [35,36].

Recently, infrastructure construction has been regarded as an important driver of urban sprawl, with the transformation of urbanization from high-speed to high-quality development [8]. The construction of public services, communication infrastructure, and transport facilities requires large amounts of urban land, and its multiplier effect in attracting investment further drives urban sprawl [28].

## 2.2. Impact of Railway Network on Urban Sprawl

Railway networks, as a key member of transport networks, affect spatial connectivity, the flows of various factors, and functional coordination between cities [37]. Many researchers have studied the impact of railway networks on urban sprawl, but consensus has not been reached [38,39]. Railway networks mainly affects the spatial pattern of urban sprawl due to spatiotemporal compression [40], agglomeration, and diffusion [41]. Railway networks shortens the space–time distances between cities, improves regional accessibility, and reduces transportation costs [42], thereby promoting the spatial movement of production factors. The central city receives greater locational advantage benefits from the agglomeration effect, which drives its economic growth and increases the demand for land [43]. When the agglomeration reaches a certain size, the diffusion effect appears. The factors that have accumulated in the central city flow out to the peripheral cities, promoting the urban expansion of these cities [44,45]. However, some research suggests that railway networks limit the factor loss owing to the urban sprawl of cities [46]. Due to the siphon effect caused by high-speed rail, scholars found that when cities that have experienced population loss in recent years are connected to a rail network, high-speed rail exacerbates this urban shrinkage [47].

The main methods used in previous studies can be divided into two types, namely, econometric and spatial analysis methods, which are summarized in in Table 1.

**Table 1.** Main methods used in previous studies.

| Type | Method | Characteristic and Use |
|---|---|---|
| Econometric method | Multiple linear regression [48], DID [39,49], multistage DID estimation [16,47] | Focus on identifying the effects of railway opening on the extent of urban sprawl. DID taken the new railway opening as the treatment of policy implementation to examine the differences in urban sprawl before and after the treatment. |
| Spatial analysis method | Spatial morphology analysis [50], SLEUTH model [51], geographically weighted regression [46], spatial mixed logit (SML) model [52] | Focus on the spatial effect of a railway on the spatial form of urban sprawl, taking into account geographical information through ArcGIS analysis. |

*2.3. Research Gap*

Although previous studies have examined the impact of railways on urban sprawl, most of the literature has focused on the impact of railways on urban-scale change. While previous studies on this issue have mainly examined the effects of the opening of new railway stations and lines in the selected city [16,39,47,49], few of them have taken into account the different network statuses of the cities within the railway network. As China's railway system has already formed a network, the interactions of cities in the network have become more prominent. Some cities have strong railway connections with other cities, and some cities have relatively weak connections. The railway network status may play an important role in the effect of the railway on urban sprawl, but there is a lack of consideration of this point. Although some scholars analyzed the characteristics of railway networks [53], they focused on its effects on urban economic development [19], not on urban sprawl. Besides these effects, in-depth research is lacking on the detailed mechanisms of how railway networks affect urban sprawl, especially on the basis of agglomeration and diffusion theory.

In terms of research methods, previous studies have usually used the DID or multistage DID model for empirical analysis, taking the opening of new railway stations or railway lines as the treatment of policy implementation to examine the differences in urban sprawl before and after the treatment [8,16]. But, in reality, the opening of a railway in a certain period is continuous and random, that is, the treatments of policy implementation do not only occur in some specific years, so the DID or multistage DID model cannot accurately represent the changes in railway interactions between objects. Instead, as one of the mainstream methods used in complex network study areas, SNA can accurately measure the status and dynamic changes in network centrality and connection strength among the nodes in the same network, but few researches have applied SNA to study railway networks.

To fill the gap, first, we focused on the effects and mechanisms of railway network construction on urban sprawl, proposing two main hypotheses of the mechanisms through which railway networks affected urban sprawl: one is the network agglomeration mechanism, and the other is the network diffusion mechanism. Second, we applied the SNA method to measure the indicators of railway network characteristics, and we conducted panel model regression with the above indicators to verify the hypotheses regarding the network agglomeration mechanism and network diffusion mechanism using the data of 26 cities from 2011 to 2019 in the YRD in China.

**3. Hypotheses**

A literature review suggests that the construction of a railway network can affect urban land use change through spatiotemporal [12], agglomeration [47,54] and diffusion effects [55,56]. In order to more clearly illustrate the spatial effects of railway networks on urban sprawl and their mechanisms, we established the analytical framework shown in Figure 1. Due to the spatiotemporal effect, a railway network can effectively shorten

the spatiotemporal distance, improve the spatial accessibility and connectivity between cities, and expand the range of economic activities, thus facilitating the spatial flow of resources and factors. Due to the combined effects of agglomeration and diffusion forces, railway networks drive the spatial agglomeration and diffusion of factors, which cause urban sprawl in the agglomeration and diffusion patterns. Hypotheses were put forward on the basis of this framework.

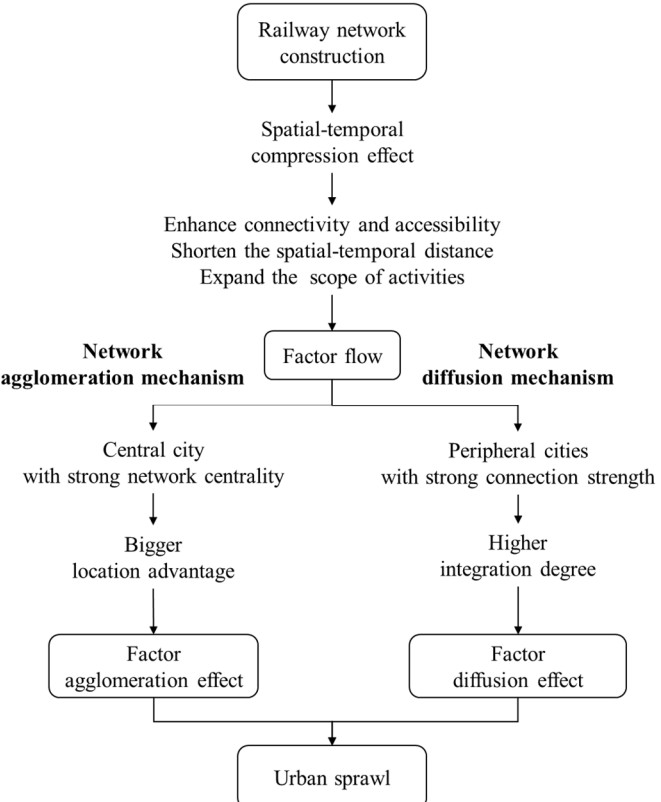

**Figure 1.** The analytical framework between a railway network and urban sprawl.

### 3.1. H1: Mechanisms of Network Agglomeration and Urban Sprawl

The network location condition refers to the location of a node city in a railway network and its spatial connection with other nodes. Figure 2 shows the location condition of cities in the railway network in an urban agglomeration. $A_i$ denotes a central city, $B_i$ denotes a secondary central city, and $C_i$ denotes a peripheral city. The construction of a railway network can significantly improve the location conditions of cities in the network by strengthening the accessibility and frequency of connections with other node cities. Through the agglomeration effect, central cities with greater locational advantages can attract the inflow of population, industry, and capital from peripheral cities and promote their spatial concentration under an agglomeration force, which drives the demand for urban space for human economic activities in central cities. Therefore, we propose the following hypothesis:

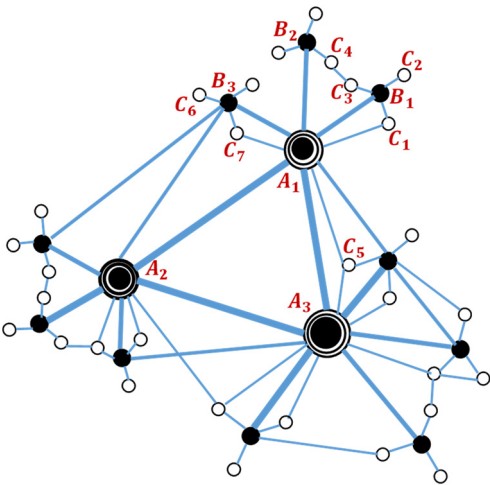

**Figure 2.** Schematic diagram of the location condition of cities in a railway network.

**Hypothesis 1.** *Network agglomeration mechanism encourages urban sprawl in the agglomeration pattern by improving the location condition of the central cities on the railway network.*

### 3.2. H2: Mechanism of Network Diffusion and Urban Sprawl

The degree of network integration reflects the status of cities in metropolitan areas in terms of infrastructure interconnection, industry convergence, market integration, resource and factor sharing, and cooperation across administrative boundaries. Figure 3 shows the different degrees of integration of peripheral cities with the central city. $A_i$ denotes the central city, $B_i$ denotes a secondary central city, and $C_i$ is a peripheral city.

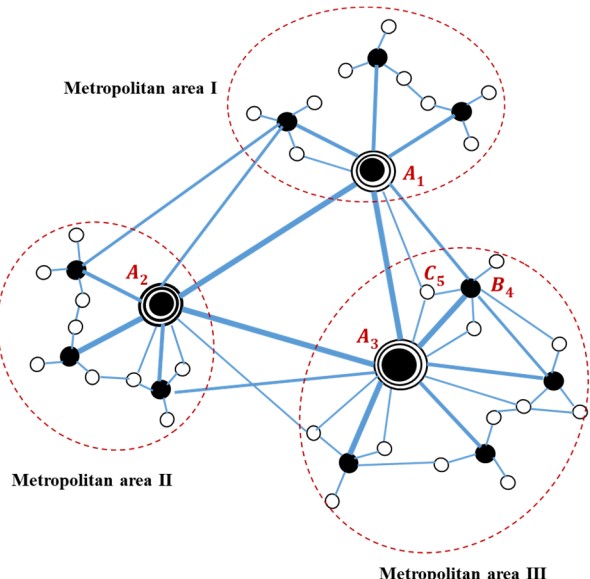

**Figure 3.** Schematic diagram of the integration degree of the peripheral cities with the central city.

The construction of a railway network can greatly improve the degree of integration of peripheral cities with a central city by strengthening the connection between the node cities in the railway network. Because of the diffusion effect, the peripheral cities with a higher integration degree can attract and benefit from the outflow of population, capital, technology, and industry from the central city, and promote their spatial diffusion on the network under the diffusion force, which drives the spatial demand in the peripheral cities. Therefore, we hypothesizes the following:

**Hypothesis 2.** *Network diffusion mechanism encourages urban sprawl in the diffusion pattern by enhancing the integration degree of the peripheral cities with the central city in the railway network.*

## 4. Materials and Methods

### 4.1. Study Area

#### 4.1.1. Yangtze River Delta Urban Agglomeration (YRD UA)

The YRD UA is one of the regions with the most complete industrial system, the fastest economic development, and the strongest comprehensive strength in China, characterized by a dense population, developed economy, and scarce land resources. For more than 40 years after China's reform and opening up, the economic development and urbanization process in the YRD UA has been progressing steadily. The regional GDP exceeded CNY 29 trillion in 2022, accounting for nearly a quarter of China's total GDP [57]. The permanent population of the YRD UA was nearly 237 million in 2022, and the urbanization rate had reached over 60% [19], ranking first among all urban agglomerations in China. The YRD UA covers an area of 217,700 square kilometers, about 2.2% of China, including 36,000 square kilometers of built-up area [58]. The YRD UA currently includes 27 cities in the provinces of Shanghai, Jiangsu, Zhejiang, and Anhui [59]. A diagram of the YRD UA is shown in Figure 4. As Zhoushan in Zhejiang province still did not have a railway station at the end of 2020, we considered the other 26 cities in the YRD UA except for Zhoushan as the study subject.

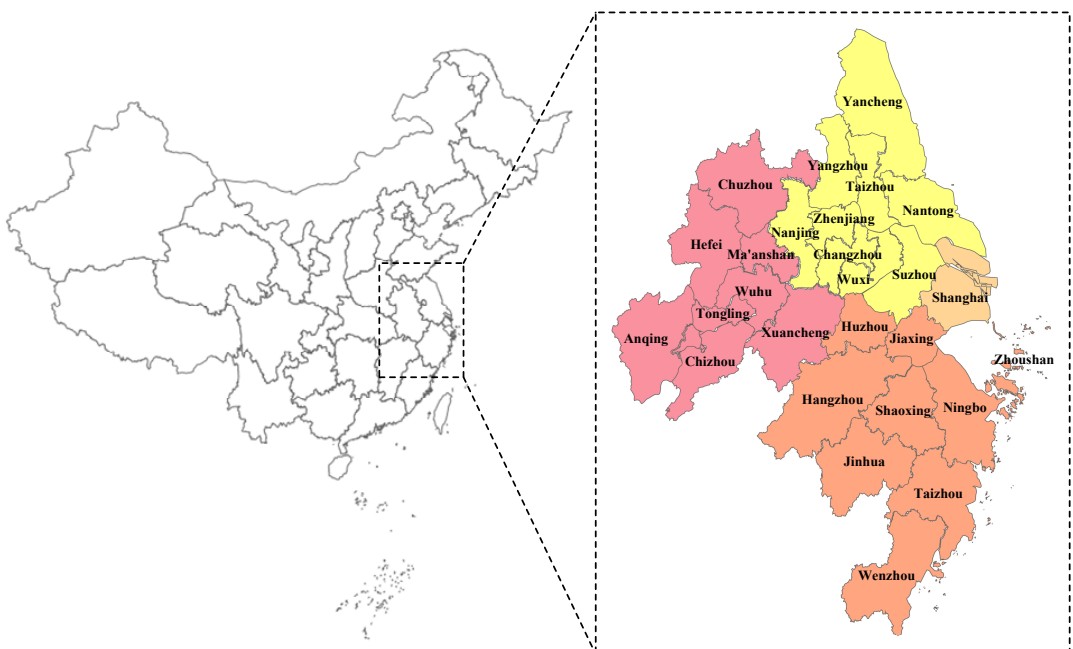

**Figure 4.** Diagram of the Yangtze River Delta Urban Agglomeration (YRD UA).

#### 4.1.2. Railway Network Construction in the YRD UA

In 2008, the first high-speed railway in the YRD UA was put into operation, officially ushering the region into the high-speed-railway era [60]. The YRD region has built a comprehensive intercity network of high-speed rail and expressways, making it possible to travel between central cities in 1.5 h [59]. By the end of 2020, there were 24 high-speed railway lines in operation in the YRD UA, with the length of the operating railway lines exceeding 12,800 km, and the length of operating high-speed-railway lines exceeding 6000 km, according to statistics from the Railway Administration [61]. At present, the construction of the railway network in the YRD UA has been completed, except for in Zhoushan in Zhejiang province. Prefecture-level cities have been connected by high-speed railway and are integrated into the "half-hour to three-hour economic circle". In

addition, the YRD is expected to build the "Yangtze River Delta on track" by 2025, forming a multilevel rail transport system consisting of long-distance railways, intercity railways, urban railways, and rail transit [19].

In terms of spatial layout, the railway network in the YRD UA has a topological spatial structure with a dense center and a sparse periphery. As the core of the network, the cities along the Shanghai–Nanjing–Suzhou–Wuxi–Changzhou railway line and the Shanghai–Jiaxing–Hangzhou railway line are closely connected. The railway network in the YRD UA basically forms a network pattern with Shanghai and Nanjing as the radiating center and Suzhou, Wuxi, Changzhou, and Hangzhou as the main supporting points.

### 4.1.3. Urban Sprawl in the YRD UA

Shanghai, Nanjing, Hangzhou, and Hefei are the main economic centers of the YRD UA, of which the total built-up area accounted for 44% of the YRD region in 2019; and with the addition of the six secondary central cities Suzhou, Wuxi, Changzhou, Ningbo, Wenzhou, and Shaoxing, the total share reached 71%. It is obvious that the spatial layout shows the characteristics of agglomeration. With the rapid development of the economy and the continuous improvement in infrastructure construction, the cities in the YRD UA have continued to expand. Figure 5 shows the average annual increase and growth rate of the 26 cities from 2011 to 2019. In terms of the average annual growth of the urban built-up area, the five cities with the highest growth rates were Shanghai, Hangzhou, Nanjing, Hefei, and Suzhou, all of which were core cities. In terms of average annual growth rates, those of Shanghai and Nanjing were lower among the four core cities, only 2.72% and 3.24%, respectively, while Hangzhou and Hefei had higher growth rates of 5.18% and 4.45%, respectively. In particular, small and medium cities such as Changzhou, Jiaxing, and Taizhou led the way, with growth rates of over 5.00%. Overall, urban sprawl in the YRD UA can be seen in both central and peripheral cities.

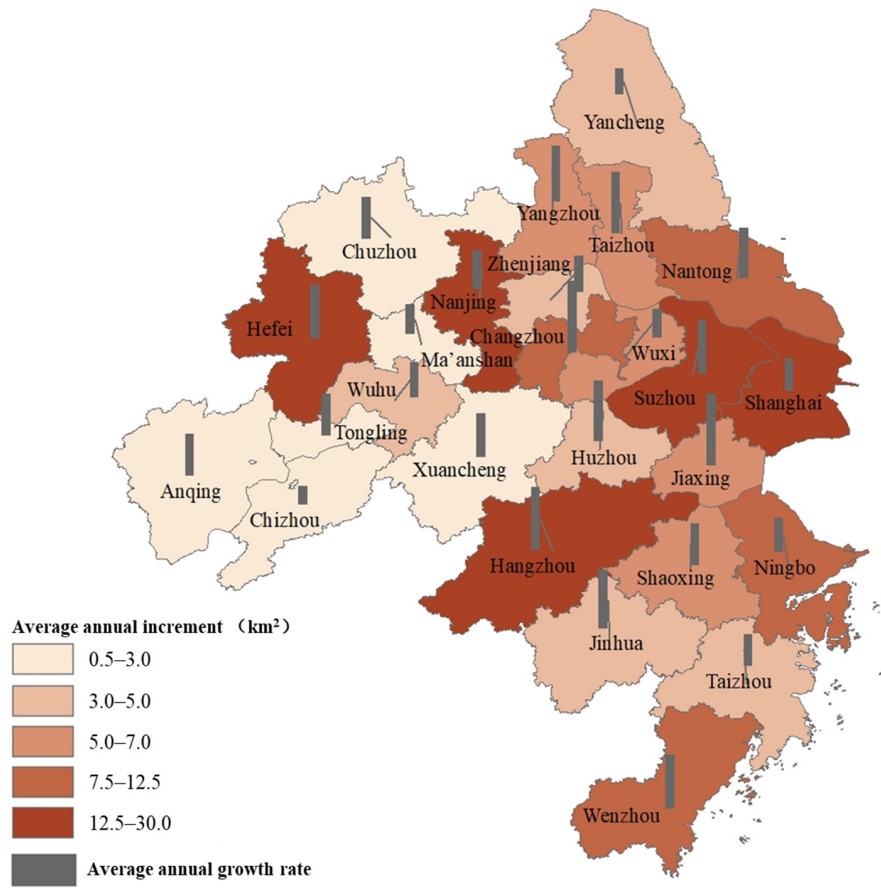

**Figure 5.** Average annual increase and growth rate of the built-up areas of 26 cities from 2011 to 2019.

*4.2. Data Source and Variables*

4.2.1. Data Source

We collected a dataset of relevant data for the 26 cities in the YRD UA from 2011 to 2019. Table 2 shows the data sources in detail.

**Table 2.** Data sources.

| Data | Source | Detailed Data Source Demonstration |
|---|---|---|
| Land | China Urban Construction Statistical Yearbook 2011–2019, published by the Ministry of Housing and Urban-Rural Development of China. | The urban built-up area data were collected from the "built-up area" column in Table 3 of the urban population and built-up area of the yearbook. |
| Railway network centrality and connection strength | National Railway Passenger Train TimeTable 2011–2019, published by the State Railway Administration of China. | Calculated with the railway train operation and daily frequency data collected from the columns "Train number", "Departure station", "Terminal station", "Transfer station" in the train timetable. |
| Population size | City Statistical Yearbook 2011–2019, published by the statistical bureau of each city. | Column "Size of permanent population" in the chapter "Population, employment and wages". |
| GDP | | Column "Gross domestic product" in the chapter "National accounts". |
| Industrial structure | | Column "GDP of secondary industries", "GDP of tertiary industries" in the chapter "National economic accounts". |
| Educational condition | | Column "Number of students per 10 000 inhabitants" in the chapter "Education". |
| Medical condition | | Column "Number of hospital beds per 10 000 inhabitants" in the chapter "Health, social protection and social assistance". |
| Ecological condition | | Column "Green cover of built-up area" in the chapter "Urban development". |
| Land resource endowment | China Urban Construction Statistical Yearbook 2011–2019, published by the Ministry of Housing and Urban-Rural Development of China. | Column "Urban district area" in Table 3, Urban population and built up area of the Yearbook. |
| Road condition | | Column "Area of urban construction land for roads, streets and transport" in Table 3 Urban Population and Construction Land of the Yearbook. |

**Table 3.** Definitions and descriptive statistics of the variables.

| Variable | Definition | Mean | SD | Min | Max |
|---|---|---|---|---|---|
| Land | The urban built-up area | 238.98 | 242.23 | 35.62 | 1237.85 |
| Degree | The degree centrality of railway network | 0.2312 | 0.2277 | 0.0000 | 1.0000 |
| Corestr | The railway connection strength with central city | 91.61 | 72.42 | 0.00 | 268.00 |
| Population | Permanent population | 615.40 | 434.06 | 73.13 | 2428.00 |
| GDP | GDP | 5588.54 | 5675.76 | 372.49 | 38155.32 |
| Industrial | Share of secondary and tertiary industries in GDP | 94.23 | 4.21 | 79.60 | 99.70 |
| Education | Number of students per 10,000 inhabitants | 1309.67 | 261.02 | 812.79 | 1962.09 |
| Hospital | Number of hospital beds per 10,000 inhabitants | 44.94 | 9.80 | 24.33 | 77.18 |
| Resources | The urban area | 2271.97 | 1646.25 | 340.00 | 8006.23 |
| Ecocondition | Green cover of built-up area | 41.93 | 2.81 | 22.26 | 48.50 |
| Urbanroad | Road area per capita | 22.22 | 7.29 | 4.04 | 46.40 |

4.2.2. Variables

This section presents the definitions and descriptive statistics of each variable in the panel regression, as shown in Table 3.

(1) Dependent variable ($Y_{it}$): $Land_{it}$ is the urban built-up area, an indicator measuring the size of urban land.

(2) Independent variable ($X_{it}$): measured using SNA.

First, we chose network centrality as a proxy variable for the location condition of the object city. The indicator of network centrality, degree centrality ($Degree_{it}$), was used to examine the effect of the agglomeration mechanism of the railway network on urban sprawl. Then, connection strength was chosen as a proxy variable for the integration degree of the target city. The integration degree indicator, connection strength with the central city ($Corestr_{it}$), was used to verify the impact of the railway network diffusion mechanism on urban sprawl.

SNA has been proven to be a powerful quantitative method for measuring the accessibility and connectivity of the nodes in a network [62]. In this study, we used SNA to measure the 4 indicators of the independent variables mentioned above. Specifically, degree centrality ($Degree_{it}$) measures the location condition of an object in a network in terms of its connections with its neighbors, calculated using Equation (1). Connection strength measures the integration degree of the object in a network in terms of the total number of connections and the frequency of connections, calculated using Equation (2).

$$C_{RDi} = C_{ADi}/(N-1) \tag{1}$$

where $C_{RDi}$ denotes the relative degree centrality of node $i$; $C_{ADi}$ is the absolute degree centrality.

$$S_i = \sum_{j=1}^{n} S_{ij} \tag{2}$$

where $S_i$ denotes the total connection strength of node $i$; $S_{ij}$ shows the connection strength between node $i$ and node $j$.

(3) Control variable ($Z_{it}$)

To ensure the validity of panel regression, we included some control variables in the model. According to relevant studies, population, economy, public services, policies, resources and environment, and traffic conditions are the main factors influencing urban sprawl [16,32]. We selected population size, economic volume, industrial structure, educational condition, medical condition, land resource endowment, ecological condition, and road condition as the control variables.

*4.3. Model Specification*

Considering that changes in urban land area are influenced by various factors and given the need to include the control factors, we set up a panel regression model as the baseline model in Equation (3):

$$Y_{it} = \alpha_0 + \alpha_1 X_{it} + \gamma \sum_n Z_{it} + \mu_i + \varepsilon_{it} \tag{3}$$

where $Y_{it}$ is the dependent variable. $X_{it}$ are the independent variables, and $\sum_n Z_{it}$ are the control variables. $\alpha_0$ is the constant term, $\alpha_1$ is the regression coefficient of the core explanatory variable. $\gamma$ is the regression coefficient of the control variable, $\mu_i$ is the unobservable individual effect, and $\varepsilon_{it}$ is the individual and time-varying disturbance term.

*4.4. Estimation Strategies*

First, we conducted the baseline model regression. To estimate the effect of the network agglomeration mechanism of the railway network on urban sprawl (Hypothesis 1), we included the proxy indicator of local conditions: network degree centrality ($Degree_{it}$) in model M1. Equation (3) was transformed into Equation (4).

$$M1: Land_{it} = \alpha_0 + \alpha_1 Degree_{it} + \gamma \sum_n Z_{it} + \mu_i + \varepsilon_{it} \tag{4}$$

To estimate the impact of the network diffusion mechanism on urban sprawl (Hypothesis 2), we included the proxy indicator of connection strength: central city connection

strength ($Corestr_{it}$) in model M2, then conducted the panel regression. Equation (3) was transformed into Equation (5).

$$M2 : Land_{it} = \alpha_0 + \alpha_1 Corestr_{it} + \gamma \sum\nolimits_n Z_{it} + \mu_i + \varepsilon_{it} \tag{5}$$

Second, we further performed heterogeneity analysis by dividing the sample into 3 groups to investigate the heterogeneity of the mechanisms.

Third, we performed robustness checks of the baseline model by substituting the core explanatory variables and running a subsample regression.

## 5. Empirical Results

### 5.1. Baseline Regression Results

In order to ensure the validity of our panel regression, we needed to determine the specific type of panel model. In practice, it is difficult to meet the conditions of the random-effects model, whereas the fixed-effects model provides consistent estimates regardless of the hypothesis, and its results are relatively more robust [63]. For the sake of rigor, we further tested it using the Hausman test, which showed a *p*-value of 0.000. Therefore, we chose the fixed-effects model. We then tested the significance of the individual and time effects, and the results showed that the individual effects were significant, while the time effects were not. In summary, we conducted empirical research using the individual fixed-effects model. The baseline regression results are shown in Table 4.

**Table 4.** The results of baseline regression.

| Variable | M1 | M2 |
|:---|:---:|:---:|
| Degree | 103.582 *** | |
| | (6.65) | |
| Corestr | | 0.102 *** |
| | | (5.53) |
| Population | 0.150 ** | −0.359 *** |
| | (2.07) | (−3.48) |
| GDP | 0.009 *** | 0.014 *** |
| | (13.20) | (14.22) |
| Industrial | −1.584 ** | −0.412 |
| | (−2.07) | (−0.61) |
| Education | −0.013 ** | −0.010 ** |
| | (−2.24) | (−2.03) |
| Hospital | 0.942 *** | 0.451 *** |
| | (5.31) | (2.67) |
| Resources | 0.005 *** | 0.004 *** |
| | (4.69) | (3.27) |
| Ecocondition | 0.372 | 0.699 ** |
| | (0.92) | (1.98) |
| Urbanroad | 1.076 *** | 0.768 ** |
| | (3.09) | (2.56) |
| Constant | 146.392 ** | 236.701 *** |
| | (2.10) | (3.84) |
| Individual fixed effects | Yes | Yes |
| Observations | 234 | 198 |
| R-squared | 0.934 | 0.910 |

Notes: ** and *** denote the significance levels of 0.05 and 0.01, respectively. The value in the parentheses is the *t* value.

### 5.1.1. The Effect of Network Agglomeration Mechanism on Urban Sprawl

In Table 4, column "M1" shows the estimation results of the effect of the railway network agglomeration mechanism on urban sprawl (see Hypothesis 1). Network centrality was the proxy variable for the location condition of the city in the railway network. The coefficient of its indicator degree centrality ($Degree_{it}$) was positive at the significant level

of 1%. Specifically, for every 1% increase in network centrality, the urban built-up area increased by 100%. This indicates that the improvement in the location condition of the city on the railway network has a positive effect on urban sprawl by strengthening the network centrality.

The construction of a railway network can effectively improve the location condition of a city by increasing railway connectivity and frequency, and shortening the spatiotemporal distance with other cities [64]. According to agglomeration theory, a city with greater locational advantages benefits from the agglomeration effect; that is, the city with high centrality can attract the inflow of resources and production factors from the peripheral cities, which promote the spatial concentration of people and economic activities through the agglomeration force [65]. In this process, the demand for construction land in the central city of the agglomeration constantly increases. This is exactly how the network agglomeration mechanism affects urban sprawl.

To further illustrate the network agglomeration mechanism, we analyzed the scale of urban sprawl in cities with different location conditions from 2011 to 2019, which were divided into three groups according to the degree centrality of the railway network, as shown in Table 5. The nine cities with high network centrality accounted for 63.9% of the total growth in the YRD UA, while the cities with medium and low network centrality accounted for only 18.8% and 17.3%, respectively. The average area growth rate of high-network-centrality cities reached 7.1%, which is 3.1 times that of medium-network-centrality cities and 3.7 times that of low-network-centrality cities. This shows that urban sprawl was mainly concentrated in cities with high rail network centrality in the period of 2011–2019.

**Table 5.** Urban sprawl scale of cities with different network centrality levels.

| Network Centrality Level | Cities | Growth in Urban Built-Up Area | | |
| --- | --- | --- | --- | --- |
| | | Total Growth (km²) | Total Proportion | Average Proportion |
| High Level | Shanghai, Nanjing, Hangzhou, Hefei, Suzhou, Wuxi, Changzhou, Zhenjiang, Jiaxing | 1167.2 | 63.9% | 7.1% |
| Middle Level | Ningbo, Wenzhou, Shaoxing, Jinhua, Wuhu, Ma'anshan, Chuzhou, Taizhou (in Zhejiang) | 343.1 | 18.8% | 2.3% |
| Low Level | Huzhou, Chizhou, Taizhou (in Jiangsu), Nantong, Yancheng, Anqing, Yangzhou, Tongling, Xuancheng | 315.3 | 17.3% | 1.9% |

In summary, our results support Hypothesis 1. The network agglomeration mechanism promotes urban sprawl in the agglomeration pattern by improving the location conditions of central cities in the railway network.

### 5.1.2. The Effect of Network Diffusion Mechanism on Urban Sprawl

In Table 4, column "M2" shows the estimation results of the effect of the railway network diffusion mechanism on urban sprawl (see Hypothesis 2). Connection strength was the proxy variable for the integration degree of the city in the railway network. The coefficient of connection strength with the central city ($Corestr_{it}$) was positive at the significance level of 1%. Specifically, for every 1% increase in connection strength, the urban built-up area increased by 0.1%. This suggests that the improvement in the strength of the connection with the central city in the railway network has a positive effect on urban sprawl.

Thanks to the construction of the railway network, the integration degree of the city in the metropolitan areas (MAs) can be greatly intensified via new operating railway lines and new railway stations. According to diffusion theory, a city with a higher degree of integration benefits from the diffusion effect; that is, a peripheral city with a higher integration degree can attract the spillover of industries, labor, and capital from the central city, which promote the spatial diffusion of people and economic activities through diffusion force [66].

In this way, the peripheral city has a growing need to expand the urban construction area for the transfer industries and economic activities. This is how the network diffusion mechanism affects urban sprawl.

To further verify the network diffusion mechanism, we calculated and compared the change in the urban built-up area of the central city and peripheral cities in the Shanghai MAs, as shown in Table 6. The urban built-up area percentage from 2011 to 2019 in Shanghai decreased by 2.6%, while those of the peripheral cities of Suzhou, Changzhou, and Jiaxing increased by 1.0%, 1.4%, and 0.8%, respectively. The peripheral cities were taking over Shanghai's industrial transfer, accelerating urban development and industrialization, and driving urban sprawl in the diffusion pattern toward the peripheral cities.

**Table 6.** The change in urban built-up area of cities in Shanghai.

|  | City | Proportion of Urban Built-up Area in Shanghai MAs | | |
|---|---|---|---|---|
|  |  | **2011** | **2019** | **Changes in Proportion** |
| Central city | Shanghai | 41.5% | 38.9% | −2.6% |
| Peripheral Cities | Suzhou | 14.0% | 15.0% | +1.0% |
|  | Wuxi | 12.0% | 10.9% | −1.1% |
|  | Ningbo | 11.8% | 11.1% | −0.7% |
|  | Changzhou | 7.2% | 8.6% | +1.4% |
|  | Jiaxing | 4.1% | 4.9% | +0.8% |
|  | Nantong | 5.9% | 6.7% | +0.8% |
|  | Huzhou | 3.5% | 3.9% | +0.4% |

In summary, the results support Hypothesis 2: the network diffusion mechanism promotes urban sprawl in the diffusion pattern by increasing the degree of integration of the peripheral cities with the central city in the railway network.

*5.2. Heterogeneity Analysis*

In this section, we use heterogeneity analysis to further explore the above-described mechanisms. Since the MAs in the YRD UA differed in terms of economic development, industry structure, and resource endowment, we divided the sample into three groups and conducted the group-level regression. As shown in Table 7, column "All", for which we used the baseline regression M1 as the control group, shows the test results of the connection strength of each city with the four central cities of Shanghai, Nanjing, Hangzhou, and Hefei in the whole YRD UA. Columns "Group 1", "Group 2", and "Group 3" represent the connection strength of each city only with Shanghai in the Shanghai MA, only with Nanjing in the Nanjing MA, and only with Hangzhou in the Hangzhou MA, respectively.

**Table 7.** The results of heterogeneity analysis.

|  | M1 | M3 | | |
|---|---|---|---|---|
| Variable | All (YRD UA) | Group 1 (Shanghai MA) | Group 2 (Nanjing MA) | Group 3 (Hangzhou MA) |
| Corestr | 0.102 *** | 0.227 *** | 0.057 | 0.180 |
|  | (5.53) | (4.18) | (0.63) | (1.46) |
| Population | −0.359 *** | −0.425 *** | 0.442 ** | −0.987 *** |
|  | (−3.48) | (−4.73) | (2.01) | (−2.83) |
| GDP | 0.014 *** | 0.013 *** | 0.018 *** | 0.024 *** |
|  | (14.22) | (9.58) | (9.81) | (4.23) |
| Industrial | −0.412 | 1.070 | −1.522 ** | −3.728 |
|  | (−0.61) | (0.66) | (−2.22) | (−0.63) |
| Education | −0.010 ** | −0.049 *** | −0.020 *** | −0.100 ** |
|  | (−2.03) | (−3.23) | (−2.89) | (−2.65) |

**Table 7.** *Cont.*

| Variable | M1 | M3 | | |
| | All (YRD UA) | Group 1 (Shanghai MA) | Group 2 (Nanjing MA) | Group 3 (Hangzhou MA) |
|---|---|---|---|---|
| Hospital | 0.451 *** | 1.232 *** | 0.136 | −0.027 |
| | (2.67) | (4.94) | (0.45) | (−0.05) |
| Resources | 0.004 *** | 0.001 | 0.016 | 0.009 *** |
| | (3.27) | (0.50) | (1.64) | (2.87) |
| Ecocondition | 0.699 ** | −0.055 | −0.269 | 0.699 |
| | (1.98) | (−0.08) | (−0.40) | (0.56) |
| Urbanroad | 0.768 ** | −0.558 | 0.352 | 2.428 ** |
| | (2.56) | (−1.39) | (1.23) | (2.13) |
| Constant | 236.701 *** | 300.267 | 60.380 | 946.277 |
| | (3.84) | (1.60) | (0.63) | (1.54) |
| Individual fixed effects | Yes | Yes | Yes | Yes |
| Observations | 198 | 63 | 63 | 45 |
| R-squared | 0.910 | 0.985 | 0.968 | 0.920 |

Notes: ** and *** denote the significance levels of 0.05 and 0.01, respectively. The value in the parentheses is the t value.

These results suggest that the network diffusion mechanism was heterogeneous in different MAs. Specifically, in the Shanghai MA (Group 1), the coefficient of connection strength ($Corestr_{it}$) was positive and passed the significance test at the 1% level. For every 1% increase in the connection strength between node cities and Shanghai, the urban built-up area increased by 0.2%. However, we did not find the same effect in the other two MAs. In the Nanjing MA (Group 2) and Hangzhou MA (Group 3), the coeffects of $Corestr_{it}$ did not pass the significance test. This indicates that the radiation and driving power of the central city to the peripheral cities of the Nanjing MA and Hangzhou MA had not yet formed, so the network diffusion effect was still weak.

We then further investigated the influence of heterogeneity. According to agglomeration and diffusion theory, the diffusion effect in MAs is related to the agglomeration level of the central city. This means that central cities with a high agglomeration level can have a strong radiating and driving ability to strengthen peripheral cities and make the network diffusion mechanism work. To further support this view, we analyzed the agglomeration level of the central city in terms of GDP, land area, and population, and the change in land area of the peripheral cities in the Shanghai, Nanjing, and Hangzhou MAs, as shown in Table 8.

**Table 8.** Central city's agglomeration level and peripheral cities' land area change.

| MAs | | Agglomeration Level of Central City (Proportion in Urban Agglomeration) | | | Changes in Urban Built-Up Area Proportion |
| | | GDP | Urban Built-Up Area | Resident Population | |
|---|---|---|---|---|---|
| Shanghai MA | Central city (Shanghai) | 19.2% | 19.1% | 15.0% | |
| | Surrounding cities | | | | +2.6% |
| Nanjing MA | Central city (Nanjing) | 6.1% | 12.2% | 5.2% | |
| | Surrounding cities | | | | +1.0% |
| Hangzhou MA | Central city (Hangzhou) | 7.0% | 8.3% | 5.6% | |
| | Surrounding cities | | | | −1.7% |

It can be seen that the agglomeration level of Shanghai was significantly higher than that of Nanjing and Hangzhou, and the proportion of built-up areas in the peripheral cities of Shanghai, Nanjing, and Hangzhou changed by +2.6%, +1.0%, and −1.7%, respectively. The time, only Shanghai MA reached a high level of agglomeration and urban integration,

which were driving the urban sprawl in the peripheral cities. Meanwhile, the degree of urban integration was not as high in the Nanjing MA and Hangzhou MA, which had relatively low agglomeration levels. This means that the diffusion effect of Nanjing and Hangzhou on the peripheral cities was weak, and they had little impact on the urban expansion of the peripheral cities.

Therefore, the heterogeneous effect of the network diffusion mechanism is related to the agglomeration level of the central city.

### 5.3. Robustness Tests

In order to verify the robustness of the above regression and findings, we replaced the core explanatory variable of degree centrality ($Degree_{it}$) of the baseline model M1 with the closeness centrality variable ($Closeness_{it}$) to build model M4 for robustness tests. As shown in Table 9, column "M4", model M4 produced similar results to model M1. This result supports the above findings that the network agglomeration mechanism promotes urban sprawl by improving the location conditions of central cities in the railway network. In detail, the closeness centrality ($Closeness_{it}$) measures the network location condition of the object in terms of the closeness to other nodes on the network, which was calculated using SNA and Equation (6).

$$C_{BPi}^{-1} = \sum_{j=1}^{n} d(i,j)/(N-1) \tag{6}$$

where $d(i,j)$ is the distance between node $i$ and node $j$; $C_{BPi}^{-1}$ is the relative closeness centrality.

**Table 9.** The results of robustness tests for baseline model.

| Variables | M4: Robustness Test of M1 | M5: Robustness Test of M2 | |
| --- | --- | --- | --- |
| | | Subsample 1 | Subsample 2 |
| Closeness | 39.945 ** | | |
| | (2.04) | | |
| Metrostr | | 0.073 *** | 0.133 *** |
| | | (4.65) | (2.76) |
| Population | 0.279 *** | −0.437 *** | −1.071 *** |
| | (3.70) | (−5.00) | (−3.30) |
| GDP | 0.010 *** | 0.014 *** | 0.024 *** |
| | (16.09) | (10.15) | (4.59) |
| Industrial | −2.671 *** | 0.869 | −4.979 |
| | (−3.23) | (0.55) | (−0.95) |
| Education | −0.010 | −0.047 *** | −0.075 ** |
| | (−1.61) | (−3.25) | (−2.07) |
| Hospital | 1.170 *** | 1.197 *** | −0.075 |
| | (6.11) | (4.98) | (−0.16) |
| Resources | 0.006 *** | 0.001 | 0.008 ** |
| | (4.73) | (0.61) | (2.54) |
| Ecocondition | 0.553 | −0.473 | 0.391 |
| | (1.25) | (−0.73) | (0.34) |
| Urbanroad | 1.158 *** | −0.557 | 2.460 ** |
| | (3.00) | (−1.43) | (2.33) |
| Constant | 127.896 * | 342.828 * | 1086.481 * |
| | (1.68) | (1.88) | (1.97) |
| Individual fixed effects | Yes | Yes | Yes |
| Observations | 234 | 63 | 45 |
| R-squared | 0.921 | 0.986 | 0.931 |

Notes: *, **, and *** denote significance levels of 0.1, 0.05, and 0.01, respectively. The value in the parentheses is the t value.

Second, we replaced the core explanatory variable, connection strength, with the central city ($Corestr_{it}$) in the baseline model M2 with the connection strength with metropolitan

areas ($Metrostr_{it}$) to construct model M5, and then we resampled the observations to obtain two subsamples for robustness checks. As shown in Table 9, column "M5", both subsample 1 and subsample 2 in model M5 yielded similar results to model M2. This result supports the above findings that the network diffusion mechanism promotes urban sprawl by increasing the degree of integration of peripheral cities with the central city in the railway network.

## 6. Discussion

### 6.1. Comparison of Findings with Previous Research

This study found that the construction of a railway network has a positive influence on urban sprawl through two main mechanisms: one is the network agglomeration mechanism, which promotes urban sprawl in the agglomeration pattern by improving the location condition of central cities; the other is the network diffusion mechanism, which promotes urban sprawl in the diffusion pattern by improving the degree of integration of peripheral cities with central cities. These findings are supported by those of previous research [8,39,43], which also suggest that railways have a positive impact on urban sprawl. However, our results contrast those of some previous studies [46,47], which found that the opening of high-speed railways (HSRs) affects the urban shrinkage of cities that had experienced net out-migration in recent years. The reasons for the difference between our findings and others may be as follows:

(1) The research object. On the one hand, in this study, the research object was a railway network, including all the ordinary, express, and high-speed railways. Instead, most of the previous research has focused on HSR [8,18]. On the other hand, the previous research focused on the effects of the opening of new railway stations and railway lines [18]. This study considered the network structure and characteristics of the railway, such as the network centrality and the connectivity strength of the railway network. China's railway system has already formed a network. Some cities have strong railway interactions with other cities, and some cities have relatively weak interactions, resulting in different statuses in the railway network [67]. This plays an important role in the effect of railways on urban sprawl and should be taken into account.

(2) The model specification. Although we conducted panel regression like the previous studies, the treatment of the key factor railway in the model is different. Previously, researchers have used the DID model with the opening of new railway lines as the treatment to examine differences in urban expansion before and after policy implementation [8,18]. Instead, we applied the SNA method to construct and measure the indicators of a railway network: network centrality ($Degree_{it}$) and connection strength ($Corestr_{it}$), and we included the indictors into a FE panel model. The results can provide valuable information about the effect of a railway network, not just the effect of an individual railway station or railway line.

### 6.2. Contributions and Limitations

This study contributes to the existing research in the following aspects: On the one hand, unlike the previous ones that focused on the effect of railway construction on urban sprawl, we mainly studied the mechanism through which a railway network affects urban sprawl based on agglomeration and diffusion theory in order to reveal the path through which the railway network creates the effect, not only the effect. The results provide evidence of the network agglomeration mechanism and network diffusion mechanism on urban sprawl. On the other hand, the previous studies usually used the DID or multistage DID model for empirical analysis, and the opening of a railway was taken as the treatment of policy implementation [8,18], which has cannot accurately reflect the actual railway construction situation in practice. This study took the experience with complex network study areas as a reference, applied the SNA method to draw a picture of railway network construction and measure its proxy indicators, and included the proxy indicators in the panel regression model for empirical analysis. Instead of studying the effect of

individual railway line opening, the results provide more information on the effects of a railway network.

This study also has some limitations. We examined the impact of railway networks on urban land expansion. In fact, urban land expansion is only one aspect of urban sprawl. The signs of urban sprawl include tangible and intangible factors such as population, land, economy, society, and the ecological environment. Merely discussing urban land expansion cannot accurately evaluate the effects of railway networks on urban sprawl and their mechanisms. In addition, the study area should be expanded in future studies. The YRD UA is the most developed region in China. Considering the development differences between regions, the results may be different in other areas.

## 7. Conclusions and Implications

### 7.1. Conclusions

Urban sprawl has become a remarkable feature in the process of urbanization in China. The railway network, as a major transport infrastructure, has a profound influence on urban spatial development.

In this context, and based on agglomeration and diffusion theory, we proposed two main hypotheses regarding the mechanisms through which railway networks affect urban sprawl: one was the network agglomeration mechanism, the other was the network diffusion mechanism. We conducted panel regressions to verify them using the data from 26 cities from 2011 to 2019 in the YRD in China. The main conclusions are as follows:

(1) Railway network construction has a significant positive impact on urban sprawl. Under the spatiotemporal effect, the railway network promotes the spatial flow of resources and factors. Under agglomeration and diffusion forces, resources and factors concentrate in central cities, and, at some stage, they diffuse from the central cities to the peripheral cities, which gives rise to the network agglomeration mechanism and network diffusion mechanism of urban sprawl.

(2) The network agglomeration mechanism improves the location condition of the central cities in the railway network, which promotes urban sprawl in the agglomeration pattern. Under the driving force of agglomeration, the central cities with greater location advantages can attract the inflow of resources and factors from the peripheral cities and promote their spatial agglomeration. By this way, the increasing human economic activities cause urban sprawl in the central cities.

(3) The network diffusion mechanism enhances the integration of peripheral cities with the central city in the railway network, which promotes urban sprawl in the diffusion pattern. Under the driving force of diffusion, the peripheral cities with a higher degree of integration can attract the spillover of resources, factors, and industries from the central cities and promote their spatial diffusion on the network, which drives urban sprawl in the peripheral cities. Moreover, the network diffusion mechanism is heterogeneous across MAs. Only the MAs with a central city that has a high degree of agglomeration have a strong diffusion effect in promoting the urban sprawl of their peripheral cities, while other MAs do not.

### 7.2. Implications

These findings have the following implications for policy makers:

(1) National land planning and urban spatial planning should be made scientific by considering the network structure of urban areas and promoting the spatial distribution of land in the multicenter and network pattern. The strategic positioning and development priorities of different cities should be clarified based on the locational advantages of central cities and the development conditions of peripheral cities. The spatial agglomeration and diffusion advantages of each city should be optimized to avoid uncontrolled urban sprawl.

(2) Urban agglomeration development plans should be scientific. On the one hand, the regional central city should be vigorously developed to promote the concentration of

production factors, industries, and capital. On the other hand, we must rely on comprehensive transport networks, strengthen economic links, and complementary functions among peripheral cities to ensure regional balance.

(3) The transport infrastructure system should be improved by building a modern transport infrastructure system that is highly interconnected and networked, so as to optimize the network agglomeration and network diffusion effects of transport infrastructure and to promote the spatial flow and efficient allocation of resources and factors.

**Author Contributions:** Conceptualization, Y.Y. and F.H.; data curation, Y.S.; formal analysis, F.H.; investigation, Y.Y.; methodology, Y.Y. and Y.S.; software, Y.S.; validation, Y.Y.; visualization, Y.S.; supervision F.H.; writing—original draft preparation, Y.Y., F.H. and Y.S.; writing—review and editing, Y.Y. and Y.S. All authors have read and agreed to the published version of the manuscript.

**Funding:** This research did not receive any specific grant from funding agencies in the public, commercial, or not-for-profit sectors.

**Data Availability Statement:** The data are available on request.

**Conflicts of Interest:** The authors have no conflicts of interest to declare.

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
