# Peer review of "Spatial Effects of Railway Network Construction on Urban Sprawl and Its Mechanisms: Evidence from Yangtze River Delta Urban Agglomeration, China"

_land, doi:10.3390/land13010025_

Round 1

Reviewer 1 Report

Comments and Suggestions for Authors

This article falls within the scope of the Land, but there are some aspects that need further improvement:

1.        In the first paragraph of the introduction section, the authors should provide some data to reflect the trend of urban sprawl.

2.        The structure of the introduction is disorganized, with some paragraphs being too short and others too long. Additionally, the coherence and cohesion between paragraphs appears unclear.

3.        The literature review section is not comprehensive. Consider focusing on and citing the following references.

          How urban sprawl influences eco-environmental quality: Empirical research in China by using the Spatial Durbin model. Ecological Indicators

          Does smart city policy promote urban green and low-carbon development?. Journal of Cleaner Production

          The effects of urbanization and urban sprawl on CO2 emissions in China. Environment, Development and Sustainability

          Exploring the temporal and spatial effects of city size on regional economic integration: Evidence from the Yangtze River Economic Belt in China. Land Use Policy

4.        The subheading of the research review is somewhat ambiguous. Consider changing it to "research gap" and highlight the novelty of your article.

5.        Section 3.1 is too simplistic. Consider adding more information and references.

6.        Could Table 1 be presented in the form of a graph?

7.        The presentation of Figure 5 is not appropriate. Consider changing it to a table for better reporting. The authors should review more literature to learn how to present their results.

8.        The first half of section 5 is somewhat redundant. The authors should integrate and condense the content in this section.

9.        Table 4 and Figure 6 have similar issues as mentioned in comment 7.

10.     In section 6, the authors present many results, but the exploration of these results is not in-depth, and they do not effectively highlight the crucial points of the article.

11.     The conclusions in section 7 are too numerous. Focus on 2-3 core conclusions and provide in-depth explanations.

12.     Limitations and contributions should also be discussed in the "discussion" section.

13.     The authors should consider citing more recent articles (published in 2021-2023). Given the research topic of the manuscript and the current reference list, I suggest focusing on the following journals: Land, Land Use Policy, Environmental Impact Assessment Review, Journal of Cleaner Production.

Author Response

We greatly appreciate your professional suggestions on the manuscript. Please see the attachment.

Reviewer 2 Report

Comments and Suggestions for Authors

Thank you for the opportunity to review the article. The article addresses the important and current theme of the relationships between transportation systems (railways) and urban growth. I find the methodology used in the article sufficient to support the hypotheses. This methodology has also adequately supported the presented results, conclusions, and formulated policy implications. As a reviewer, I would like to raise the following reservations about the content of the article:

·         A good practice in scientific articles is to clearly and distinctly present hypotheses undergoing verification or research questions for which answers are sought through the study presented in the article. In the presented article, hypotheses are scattered throughout the text, with the first one appearing only on page 7.

·         The abstract contains information regarding the methodology and results. However, it lacks information regarding the main purpose of the article, namely why the authors undertook the study of the spatial effects of the railway network on urban sprawl in the Yangtze River Delta Urban Agglomeration (YRDUA) in China, using social network analysis (SNA) and a panel regression model. In this context, the statement "The spatial effect of the railway network on urban sprawl needs to be further identified both theoretically and empirically" is not sufficient.

·         In the "Research Review" section, numerous statements are made, such as "They all agreed that the railway network is the main driver of urban sprawl" or "While early studies mainly took an isolated perspective, studying the impact of railway line construction or station opening on urban sprawl of individual cities." However, not a single literary reference is cited in this section. Consequently, the reader has no opportunity to assess whether these statements are true or not.

·         In many places in the text, there is a lack of citations confirming the accuracy of numerical data. For example, the passage "The YRDUA had a population of nearly 237 million in 2022, of which the urbanization rate reached over 60%. The YRDUA's GDP exceeded CNY29 trillion in 2022, accounting for nearly a quarter of China's total GDP. The YRDUA covers an area of 217,700 square kilometers, about 2.2 percent of China, including 36,000 square kilometers for construction" is not accompanied by any citation.

Minor comments:

·    Figure 2 is illegible due to the font size being too small and the graphic elements being too tiny.

·         There are passages in the text that use a different citation style than the one applied in the majority of the article. For example, "According to the Plan for the Integrated Development of Higher Quality Transport in the Yangtze River Delta Region, issued by the National Development and Reform Commission of China, the Yangtze River Delta region has built a comprehensive intercity transport network of high-speed railways and expressways, making it possible to travel between central cities in 1.5 hours."

Author Response

(The authors gave the same response as above.)

Reviewer 3 Report

Comments and Suggestions for Authors

The title of the article indicates that the authors have chosen an important issue that deserves scientific study. It is not new because similar topics have already been published in other scientific works, but of course, I have not found the same example using a similar methodology. However, the article is very difficult to read, it is too long and too complicated. It is not read well, it will not reach a wide audience and it will not constitute an attractive source of scientific knowledge. The article requires thorough reconstruction and shortening. In particular, the description of the methodology, introduction, and summary are unclear. All these elements require fundamental change. The article was based on an extensive literature review, but it did not prove that the reviewed article complements scientific knowledge and fills the scientific gap.

Comments on the Quality of English Language

The article requires linguistic correction. Part of the scientific argument is not clear. The content requires shortening and clarifying the message. Linguistic and editing errors require correction.

Author Response

(The authors gave the same response as above.)

Reviewer 4 Report

Comments and Suggestions for Authors

The authors explore the spatial effects of railway network construction on urban sprawl. I only have a few comments.

1.    Please add a table that summarises the previous studies in this area. While your exact methodology may have never been used before, others have certainly tried to answer a similar research question. Please summarise their research in a table.
2.    And in the discussion section please evaluate whether your study supports or contradicts their results.
3.    Please provide a source for the data you mention in 5.1 Data Description. Please also exactly explain what these datasets look like (e.g. columns of the dataset).
4.    I found it rather strange to find the results of whether the hypotheses are supported or rejected before the result section in section 5. Data and Methodology.

Author Response

(The authors gave the same response as above.)

Round 2

Reviewer 1 Report

Comments and Suggestions for Authors

The authors have addressed all my concerns.